# Mamba3D: Enhancing Local Features for 3D Point Cloud Analysis via State Space Model

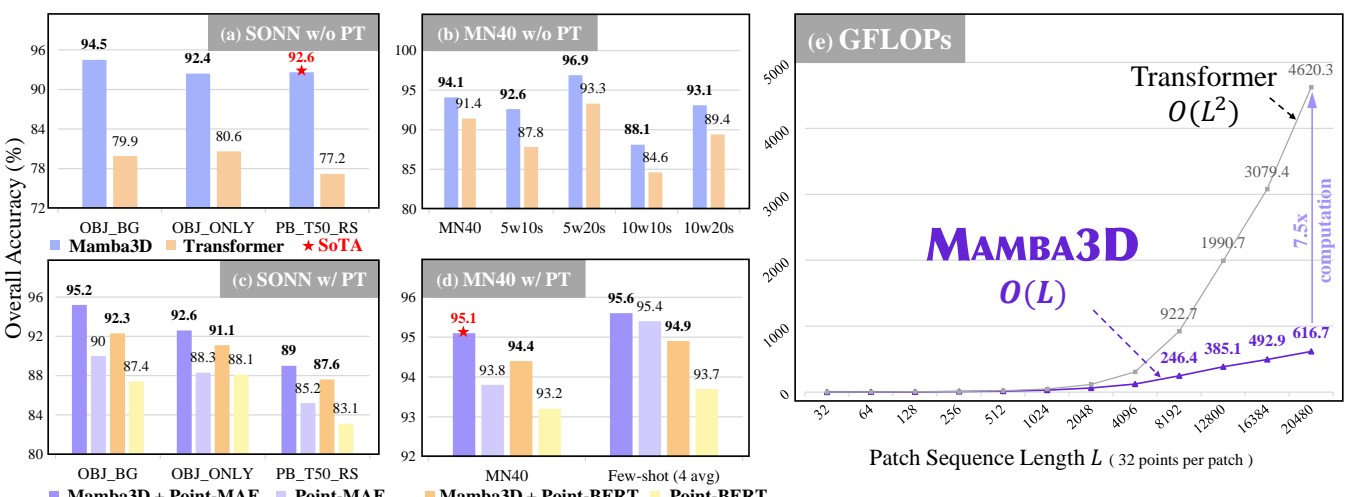

**Figure 1: Comparisons between Transformer and Mamba3D. Whether w/ or w/o pre-training (PT), Mamba3D outperforms Transformer. (a-b) Mamba3D achieves 92.6% overall accuracy (OA) on ScanObjectNN (SONN) Classification task, setting new SoTA among models trained from scratch. (c-d) When using PT, our Mamba3D gets 95.1% OA on ModelNet40 (MN40) dataset, setting new SoTA among single-modal pre-trained models. Mamba3D also shows its strong few-shot learning ability on MN40. (e) Moreover, Mamba3D's FLOPs increase linearly with sequence length, whereas Transformer increases quadratically.**

## ABSTRACT

Existing Transformer-based models for point cloud analysis suffer from quadratic complexity, leading to compromised point cloud resolution and information loss. In contrast, the newly proposed Mamba model, based on state space models (SSM), outperforms Transformer in multiple areas with only linear complexity. However, the straightforward adoption of Mamba does not achieve satisfactory performance on point cloud tasks. In this work, we present Mamba3D, a state space model tailored for point cloud learning to enhance local feature extraction, achieving superior performance, high efficiency, and scalability potential. Specifically, we propose a simple yet effective Local Norm Pooling (LNP) block to extract local geometric features. Additionally, to obtain better global features, we introduce a bidirectional SSM (bi-SSM) with both a token forward SSM and a novel backward SSM that operates on the feature channel. Extensive experimental results show that

Mamba3D surpasses Transformer-based counterparts and concurrent works in multiple tasks, with or without pre-training. Notably, Mamba3D achieves multiple SoTA, including an overall accuracy of 92.6% (train from scratch) on the ScanObjectNN and 95.1% (with single-modal pre-training) on the ModelNet40 classification task, with only linear complexity. We shall release the code and model upon publication of this work.

## CCS CONCEPTS

• **Computing methodologies** → **Shape analysis**; *Point-based models*; *Computer vision.*

## KEYWORDS

Point Cloud Analysis, State Space Model, Local Feature

## 1 INTRODUCTION

3D point cloud analysis serves as the foundation of wide-ranging applications such as autonomous driving [37, 47], VR/AR [19], Robotics [45], etc. With the rich deep learning literature in 2D vision, a natural inclination is to develop deep learning methods for point cloud processing. Unlike 2D images, point clouds do not have a specific order and exhibit a complex geometric nature, which poses challenges for deep point cloud feature learning.

Starting from PointNet [35]/PointNet++ [36], deep learning on point clouds has gained popularity. A series of deep neural networks trained from scratch, such as DGCNN [52], PointMLP [31],

*ACM MM, 2024, Melbourne, Australia*
© 2024 Copyright held by the owner/author(s). Publication rights licensed to ACM.
ACM ISBN 978-x-xxxx-xxxx-x/YY/MM
https://doi.org/10.1145/nnnnnnn.nnnnnnn

PointNeXt [40] etc., are designed for robust point feature extraction. Recently, a flux of Transformer-based pre-training models [6, 10, 33, 38, 39, 61, 63] has been proposed to unleash the scalability and generalization of Transformer [50] for 3D point cloud representation learning, by leveraging a large amount of unlabelled data. However, the Transformer suffers from the dreaded quadratic bottleneck due to the pairwise communication brought by the attention mechanism. In other words, the Transformer-based model gets slower quadratically as the input size increases. Here, we focus on finding a new backbone for point cloud feature learning that achieves *superior performance, high efficiency, and scalability potential*.

The Mamba model [15], a recently proposed alternative to Transformer, is gaining attention for its efficiency. Built upon state space models (SSM), Mamba introduces a novel selection mechanism to effectively compress context, enabling it to handle long sequences. Also, the hardware-accelerated scan enables Mamba to achieve near-linear complexity during training. However, the straightforward adoption of Mamba does not achieve satisfactory performance on point cloud tasks due to the following challenges. Firstly, its recurrent/scan mode leads to sequential dependency that is unsuitable for unordered point clouds, causing unstable pseudo-order reliance. Secondly, Mamba lacks explicit local geometry extraction, which is crucial in point cloud learning [31, 34].

Driven by the above analysis, we present **Mamba3D**, a novel state space model tailored for point cloud learning. Going beyond existing works, two essential technical contributions are delivered: (1) *Local Norm Pooling (LNP)*: a local feature extraction block comprising K-norm and K-pooling operators for local feature propagation and aggregation, respectively. To ensure the efficiency and scalability of our Mamba3D, we design our LNP block as simple yet effective, utilizing only 0.3M parameters. (2) *Bidirectional-SSM (bi-SSM)*: a token forward SSM and a novel backward SSM that operates on the feature channel to obtain better global features. Considering the disorder of the point token sequence, we propose to treat the feature channel as an ordered sequence, which is more reliable and stable. Based on the original token forward (L+) SSM, we further design a feature reverse backward (C-) SSM to alleviate pseudo-order reliance, thus fully exploiting the global features.

Note that, there are concurrent works PointMamba [26] and PCM [64] that also apply Mamba to 3D point clouds. However, PointMamba ignores local feature extraction, while PCM does not support pre-training and is computationally intensive (2× parameters, 12× FLOPs). In contrast, Mamba3D yields more representative point features by explicitly incorporating local geometry. Particularly, its linear complexity and large capacity allow for both training from scratch, and equipping with various pre-training strategies, facilitating downstream tasks with promising performance.

To thoroughly evaluate Mamba3D's capacity and representation learning ability, we conduct extensive experiments by training our model from scratch, as well as pre-training using two different pre-training strategies following Point-BERT [61] and Point-MAE [33], respectively. Results show that Mamba3D substantially surpasses both the Transformer-based counterparts and the two concurrent works on various downstream tasks, while having fewer parameters and FLOPs. For example, as shown in Fig. 1(a) and Fig. 1(b), Mamba3D achieves **92.6%** overall accuracy (OA) on the ScanObjectNN [49] classification task, setting new SoTA among models

trained from scratch, and outperforms Transformer on both ModelNet40 [56] object classification task and few-shot classification task. Similarly, as shown in Fig. 1(c) and Fig. 1(d), when equipped with the pre-training strategies proposed in Point-BERT and Point-MAE, Mamba3D still outperforms Transformer on various tasks. Particularly, Mamba3D achieves **95.1%** OA on ModelNet40, setting new SoTA among single-modal pre-trained models. Meanwhile, Mamba3D reduces **30.8%** in parameters and **23.1%** in FLOPs compared to Transformer. Section 4 presents more experimental results.

In summary, this work makes the following contributions:

- We introduce Mamba3D, a state space model with local geometric features tailored for point cloud learning, achieving superior performance with linear complexity.
- We design a Local Norm Pooling (LNP) block, enhancing local geometry extraction with only 0.3M parameters.
- We propose C-SSM, a feature reverse SSM, alleviating pseudo-order reliance in unordered points.
- Extensive experiments demonstrate Mamba3D's superior performance over Transformer, achieving multiple SoTA results and robust few-shot learning capabilities.

## 2 RELATED WORK

### 2.1 Deep Point Cloud Learning

As deep neural networks (DNNs) continue to advance, point cloud feature learning has gained increasing attention, leading to the development of numerous deep architectures and models in recent years. Inspired by early models PointNet [35] and PointNet++ [36], some attempts [2, 8, 24, 25, 43, 52, 65] design various deep architectures to better capture local context information. Later, Transformer-inspired [50] models such as Point Transformer v1-v3 [54, 55, 66] and Stratified Transformer [23] have become popular backbones, integrating local and global information to achieve state-of-the-art results. However, these dedicated architectures for 3D understanding excel in specific tasks but struggle with transferring across tasks and modalities.

To fully make use of the massive unlabelled data, self-supervised pre-training thereby becomes a viable technique. For example, Point-BERT [61], Point-MAE [33], and MaskPoint [27] propose to pre-train the Transformer [50] with masked point modeling approaches [28, 48, 63]. These methods enable models to learn generalizable features, which are transferable to different tasks effectively. There are also multi-modal pre-training strategies like ACT [10], ULIP-2 [58], and ReCon [39] that leverage cross-modal information from language and images to enhance generalization and robustness. However, their Transformer-based backbones suffer from quadratic complexity, posing challenges in handling long sequences, resulting in coarse-grained patching and information loss. In contrast, Mamba3D leverages Mamba's linear complexity, surpassing Transformer in both performance and efficiency.

### 2.2 State Space Models

State Space Models (SSM) [16, 17, 22], inspired by continuous systems, have emerged as promising models for sequence modeling. Notably, S4 [16] demonstrates the ability to capture long-range dependencies with linear complexity, showcasing effectiveness across

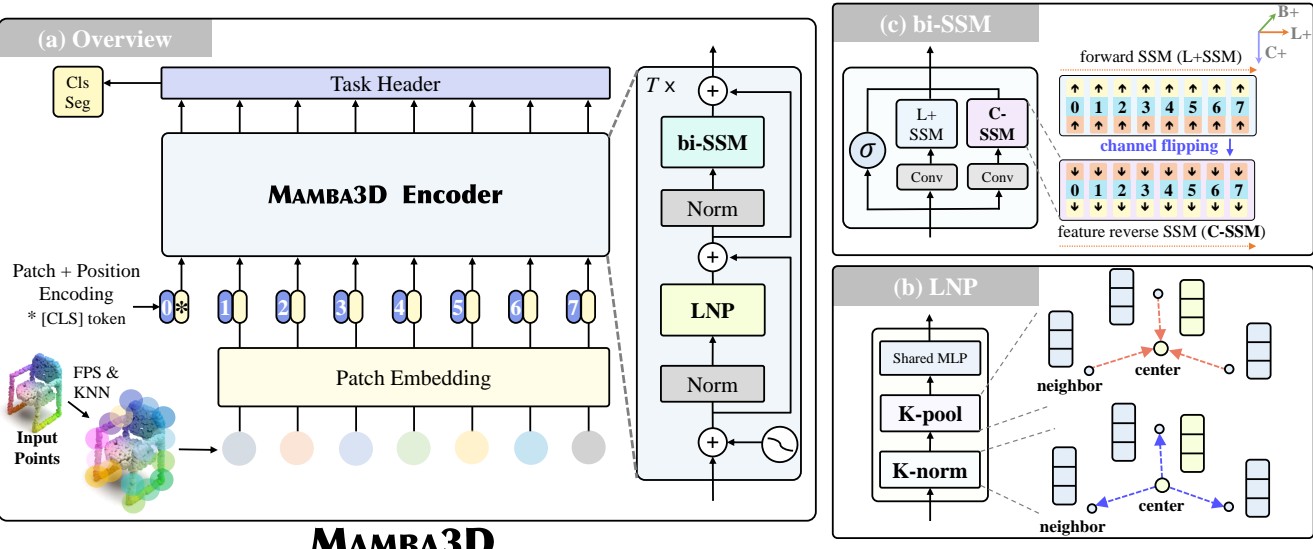

Figure 2: Illustration of Mamba3D. (a) Overview. We first segment input point cloud into $L$ patches using FPS & KNN, and then obtain the initial patch embeddings via a light-PointNet. After adding a [CLS] token, we apply standard positional encoding to all patch embeddings, which are then fed into Mamba3D Encoder. Finally we use a task header to fit downstream tasks. (b-c) Mamba3D Encoder details. The illustration of the Mamba3D Encoder is inspired by Dosovitskiy et al. [11].

diverse domains like audio [13] and vision [32]. The newly proposed Mamba model [15] further improves upon S4 by introducing a selection mechanism. By parameterizing SSM based on inputs, Mamba selectively retains relevant information, facilitating efficient processing of long-sequence data.

To adapt Mamba from sequence data to unordered point cloud data, there are concurrent works PointMamba [26] and PCM [64]. PointMamba applies Mamba directly without considering the local contexts. Based on PointMLP [31], PCM lacks pre-training and suffers from excessive parameters, limiting Mamba's efficiency. In contrast, Mamba3D effectively applies Mamba to point cloud learning, integrating the unique local geometry of point clouds. Additionally, we employ various pre-training strategies to validate Mamba3D's scalability and large capacity.

## 3 METHOD

Our aim is to leverage the Mamba model's capabilities as a backbone for point cloud feature learning, emphasizing both global receptive field and local geometric details. Below, we first briefly review the State Space Models (SSM) and the Mamba model (Section 3.1), then followed by an overview and detailed explanations of our key designs (Section 3.2-Section 3.4). Finally, we outline two pre-training strategies of our Mamba3D (Section 3.5).

### 3.1 Preliminaries: SSM and Mamba

Drawing from continuous systems, state space models (SSM) map input $x_t$ to output $y_t$ via a latent state $h_t$, with the state evolving over time $t$ continuously. In practice, to accommodate discrete data like text and images, the SSM must be discretized first:

$$h_t = \overline{\mathbf{A}}h_{t-1} + \overline{\mathbf{B}}x_t, \quad y_t = \overline{\mathbf{C}}h_t, \tag{1}$$

where $\overline{\mathbf{A}}$, $\overline{\mathbf{B}}$, and $\overline{\mathbf{C}}$ are the discrete state, control, and output matrix, respectively. Because sequential parameters $\overline{\mathbf{A}}$, $\overline{\mathbf{B}}$, and $\overline{\mathbf{C}}$ exhibit Linear Time Invariance (LTI), we can parallelize the recurrent SSM in Eq. (1) using a convolutional method into Eq. (2):

$$\overline{\mathbf{K}} = (\overline{\mathbf{CB}}, \overline{\mathbf{CAB}}, \dots, \overline{\mathbf{CA}}^{L-1}\overline{\mathbf{B}}), \quad \mathbf{y} = \mathbf{x} * \overline{\mathbf{K}}, \tag{2}$$

where $\mathsf{L}$ is the length of the input sequence $\mathbf{x}$, and $\overline{\mathbf{K}} \in \mathbb{R}^{\mathsf{L}}$ denotes a global convolution kernel, which can be efficiently pre-computed.

S4 [16] enhances SSM's ability for long sequence modeling and speed. Mamba [15] acknowledges the efficiency of LTI models like S4 in compressing extensive contexts into compact states compared to Transformer [50], which exhibits quadratic complexity during training due to zero-compression. However, the constant dynamics of LTI models, such as the input-independent parameters $\overline{\mathbf{A}}$, $\overline{\mathbf{B}}$, and $\overline{\mathbf{C}}$ in Eq. (1), limit their ability to selectively remember or forget relevant information, constraining their contextual awareness. To enhance content-aware reasoning, Mamba introduces a selection mechanism to control how information propagates or interacts along the sequence dimension. This is achieved by making the parameters that affect interactions along the sequence input-dependent, as defined in Eq. (3):

$$h_t = s_{\overline{\mathbf{A}}}(x_t)h_{t-1} + s_{\overline{\mathbf{B}}}(x_t)x_t, \quad y_t = s_{\overline{\mathbf{C}}}(x_t)h_t, \tag{3}$$

where $s_{\overline{\mathbf{A}}}(x_t)$, $s_{\overline{\mathbf{B}}}(x_t)$ and $s_{\overline{\mathbf{C}}}(x_t)$ typically denote three linear projections applied to input $x_t$. The selection mechanism addresses the limitations of LTI models but makes parallelization shown in Eq. (2) impractical. To tackle this challenge, Mamba introduces a hardware-aware selective scan to achieve near-linear complexity. Please refer to the original Mamba paper [15] for further details.

## 3.2 Mamba3D Overview

Though Mamba produces astounding results in sequential data, it is not straightforward to adapt Mamba to the 3D point cloud. On the one hand, Mamba employs recurrent/scan structure, which implies constant-time inference and linear-time training due to the effective context compression, but exhibits a unidirectional reliance. While this recurrent model works well for text, it poses challenges when dealing with unordered point clouds. On the other hand, Mamba's global receptive field cannot adequately capture local point geometry, limiting its ability to learn fine-grained features. To address the above issues, we introduce Mamba3D, featuring an effective local norm pooling (LNP) block for explicit local geometry extraction and a specialized bidirectional SSM (bi-SSM) tailored for unordered points. Fig. 2(a) shows an overview of Mamba3D.

3.2.1 **Patch Embeddings**. Given an input point cloud $\mathbf{P} \in \mathbb{R}^{N \times 3}$ with $N$ points, similar to existing works [33, 61], we first employ Farthest Point Sampling (FPS) to select $L$ central points $\mathbf{P}_C \in \mathbb{R}^{L \times 3}$. Then, for each central point $\mathbf{P}_C^i$, we construct a local patch $\mathbf{x}_p^i \in \mathbb{R}^{K \times 3}$ using $K$-Nearest Neighborhood (KNN) in $\mathbf{P}$. Finally, we employ a light-PointNet [35] to extract features for each local patch, serving as its initial patch embeddings.

3.2.2 **Mamba3D Encoder**. After obtaining the patch embeddings, they are treated as token sequences in the Transformer [50]. Similar to ViT [11] and BERT [9], we first introduce a learnable [CLS] token to aggregate information across the entire sequence. Then, we add standard learnable positional encoding [50] to these $L + 1$ tokens. The sequence is then fed into the Mamba3D encoder for high-level feature embedding, which is finally connected to a simple fully connected layer for various downstream tasks.

The design of our encoder is illustrated in the right-most part of Fig. 2(a). Specifically, inspired by MetaFormer [60], we employ a Transformer-like structure consisting of a token mixer (i.e., LNP) and a channel mixer (i.e., bi-SSM) to extract local and global features, respectively. Each block is preceded by a Layernorm [3] and followed by a residual connection [21].

Overall, the pipeline of point embedding and encoder layer is represented by the following equations:

$$\mathbf{z}_0 = [\mathbf{x}_{\text{cls}}; \mathbf{x}_p^1 \mathbf{E}; \mathbf{x}_p^2 \mathbf{E}; \cdots ; \mathbf{x}_p^L \mathbf{E}] + \mathbf{E}_{pos}, \tag{4}$$

$$\mathbf{z}'_\ell = \mathbf{LNP}(LN(\mathbf{z}_{\ell-1} + \mathbf{E}_{pos})) + \mathbf{z}_{\ell-1}, \quad \ell = 1 \ldots T \tag{5}$$

$$\mathbf{z}_\ell = \mathbf{bi\text{-}SSM}(LN(\mathbf{z}'_\ell)) + \mathbf{z}'_\ell, \quad \ell = 1 \ldots T \tag{6}$$

where $\mathbf{z}$ is the output of each layer, $\mathbf{x}_{\text{cls}} \in \mathbb{R}^{1 \times C}$ is the learnable [CLS] token, $\mathbf{E}$ is the light-PointNet to project input patches from $\mathbf{x}_p \in \mathbb{R}^{L \times K \times 3} \mapsto \mathbf{x}_p \mathbf{E} \in \mathbb{R}^{L \times C}$, and $LN$ denotes the Layernorm operation. In practice, we stack $T$ encoder layers, and a standard learnable positional encoding $\mathbf{E}_{pos} \in \mathbb{R}^{(L+1) \times C}$ is incorporated into every encoder layer, as in Point-MAE [33], to enhance the model's spatial awareness.

## 3.3 Local Norm Pooling

Local geometric features have been proven to be vital for point cloud feature learning, but were unfortunately ignored in Point-Mamba [26]. Typically, local features in point clouds are obtained by constructing a local graph using KNN, followed by feature fusion [31, 35, 36]. To ensure both effectiveness and efficiency, we design a novel Local Norm Pooling (LNP) block by simplifying the local feature extraction into two key steps: feature propagation and aggregation. Specifically, as illustrated in Fig. 2(b), LNP comprises two operators K-norm (propagation) and K-pooling (aggregation), alongside a shared MLP for channel alignment.

3.3.1 **K-norm: propagation**. After constructing a local graph with $k$ neighbors using KNN around each central point, the feature propagation involves (1) enabling neighboring points to perceive relative features concerning the central point and (2) conducting feature fusion to update their features accordingly. To achieve this, we first normalize the neighbor features $\mathbf{F}_K \in \mathbb{R}^{L \times k \times C}$ to get $\tilde{\mathbf{F}}_K \in \mathbb{R}^{L \times k \times C}$ as defined in Eq. (7). Then, we concatenate $\tilde{\mathbf{F}}_K$ with the repeated (by $K$ times) central point feature $\mathbf{F}_C \in \mathbb{R}^{L \times k \times C}$ and apply a learnable linear transformation across the local graph to obtain the propagated features $\widehat{\mathbf{F}_K} \in \mathbb{R}^{L \times K \times 2C}$:

$$\tilde{\mathbf{F}}_K = \frac{\mathbf{F}_K - \mathbf{F}_C}{\sqrt{Var(\mathbf{F}_K - \mathbf{F}_C) + \epsilon}}, \quad \widehat{\mathbf{F}_K} = [\tilde{\mathbf{F}}_K \oplus \mathbf{F}_C] * \gamma + \beta, \tag{7}$$

where $\gamma$ and $\beta$ are trainable scale and shift vectors as in Layernorm [3], respectively, and $\oplus$ signifies feature channel concatenation. This linear transformation preserves topological features while capturing the rigid transformation of the local graph features. As depicted in the lower half of Fig. 2(b), our K-norm facilitates local feature propagation from the central point to its neighbors.

3.3.2 **K-pooling: aggregation**. After propagating features within the local graph, we aggregate the information back to the central point for feature updating. While max-pooling is commonly used for feature aggregation in unordered points to maintain order invariance [35], it would lead to information loss. Inspired by Softmax, we introduce K-pooling to efficiently perform local feature aggregation while mitigating this information loss, as defined in Eq. (8):

$$\widehat{\mathbf{F}_C} = \sum_i^K \frac{\exp \widehat{\mathbf{F}_K^i}}{\sum_j^K \exp \widehat{\mathbf{F}_K^j}} \cdot \widehat{\mathbf{F}_K^i}, \tag{8}$$

where $\widehat{\mathbf{F}_C}$ is the updated central point features. K-pooling maps from $\widehat{\mathbf{F}_K} \in \mathbb{R}^{L \times k \times 2C} \mapsto \widehat{\mathbf{F}_C} \in \mathbb{R}^{L \times 2C}$, generates updated central point features, as depicted in the upper half of Fig. 2(b).

Intuitively, the LNP block constructs a local graph with adjacent patches and facilitates local feature propagation and aggregation, enabling information exchange within the local field, thus capturing the geometric and semantic features of the local patches. The receptive field of the LNP is smaller than that of SSM. By adjusting the size of the receptive field, the LNP integrates local and global information, enabling Mamba3D to more comprehensively capture the semantic information of 3D objects.

## 3.4 Bidirectional-SSM

Mamba is originally designed as a unidirectional model suited for processing 1D sequences like text. However, vision tasks often require an understanding of global spatial information. Hence, Vision Mamba [67] proposed using both forward and backward SSM to

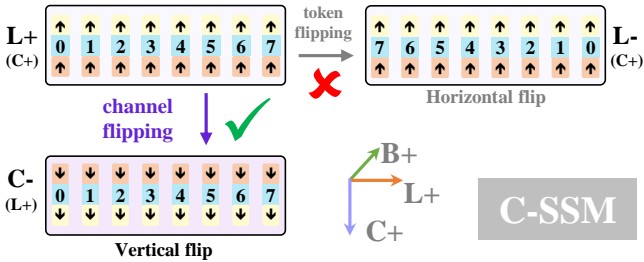

**Figure 3: Illustration of feature channel flipping. Instead of horizontal token flip, we propose a vertical feature flip, which alleviates the pseudo-order reliance.**

incorporate global information by simply horizontally flipping the token order, as illustrated by the **L+** and **L-** embeddings in Fig. 3. However, unlike the structured grid of images, point clouds are unordered and irregular, learning sequence order causes an unreliable and unstable pseudo-order dependency. To address this, we instead propose to prioritize modeling the intrinsic distribution of feature vectors rather than point tokens. Specifically, we propose a novel backward SSM, named feature reverse SSM, or C-SSM, as illustrated in Fig. 3. Combining this with the original forward SSM in Mamba, termed L+SSM, results in our bidirectional-SSM block, or bi-SSM for short. Formally, the bi-SSM block is defined as:

$$\text{bi-SSM}(\mathbf{F}) = \mathbf{F} + [\text{L+SSM}(\mathbf{F}^{L+})] + [\text{C-SSM}(\mathbf{F}^{C-})]. \quad (9)$$

With the input forward embedding **F**, also denoted as $\mathbf{F}^{L+}$, we perform a channel flip to obtain $\mathbf{F}^{C-}$, which is then fed into L+SSM and C-SSM block to generate the output embedding, respectively.

As shown in Fig. 3, we employ a vertical flip to obtain $\mathbf{F}^{C-}$, instead of a horizontal flip to get $\mathbf{F}^{L-}$. This approach reduces the pseudo-order reliance, crucial in unordered point clouds where token order lacks consistent meaning. For instance, two adjacent tokens might represent the tail and head of an airplane, respectively, which are spatially and semantically distant, posing challenges for continuous and complete feature learning. Mamba's feature selection mechanism may exacerbate this, scattering features in high-dimensional space. Instead, by reversing the feature channel, the model prioritizes learning the distribution of feature vectors. These two directions—forward token embeddings and backward feature channels—carry distinct and more reliable information, enhancing Mamba3D's ability to acquire more effective knowledge.

## 3.5 Pre-training Details

Our Mamba3D can not only be trained from scratch, but also be pre-trained with various pre-training strategies, thus facilitating downstream tasks with promising performance. In experiments, we verify the capacity and representation learning capability of Mamba3D with two commonly used pre-training strategies proposed by Point-BERT [61] and Point-MAE [33].

### 3.5.1 *Point-BERT pre-training stategy*.
Firstly, we randomly mask out 55%~85% input point embeddings, instead of a mask ratio between [0.25, 0.45] in Point-BERT. Increasing the mask ratio can not only speed up the training process, but also push Mamba3D's feature learning ability, enabling the model to learn from limited inputs. Then the Mamba3D encoder processes both visible and masked embeddings to produce a token sequence. Meanwhile, we employ the pre-trained dVAE [44] weight of Point-BERT directly to predict token sequence from point embeddings as token guidance. Lastly, we calculate the L1 loss between the encoder's output token sequence and the one from dVAE as the loss function.

### 3.5.2 *Point-MAE pre-training stategy*.
Following Point-MAE, we use a masked point modeling approach and directly reconstruct masked points. We employ an encoder-decoder architecture, where the encoder processes only visible tokens and generates their encoding. Unlike Point-MAE, our decoder employ a different architecture from encoder, containing only bi-SSM block but no LNP block, which can speed up convergence without performance loss. The encoded visible tokens and masked tokens are fed into the decoder to predict masked points. Loss is calculated using the Chamfer Distance [12] between output and ground truth points. In downstream tasks, we only use the pre-trained encoder to extract features, with task headers appended for fine-tuning.

## 4 EVALUATION

In this section, we first introduce the network implementation details. Then we evaluate Mamba3D against various existing methods in multiple downstream tasks, including object classification, part segmentation, and few-shot learning. Finally, we show the results of the ablation study for our model.

### 4.1 Implementation Details

We employ $T$=12 encoder layers with feature dimension $C$=384, and set $k$=4 in the LNP block. During pre-training, we utilize the ShapeNet dataset [4], which contains ~50K 3D CAD models covering 55 object categories. Each input point cloud, containing $N$=1024 points, is divided into 64 patches with each consisting of 32 points. Pre-training employs the AdamW optimizer [30] with cosine decay, an initial learning rate of 0.001, a weight decay of 0.05, a dropout rate of 0.1, and a batch size of 128 for 300 epochs. During fine-tuning, the point cloud is divided into 128 patches, and we train the model with the AdamW optimizer with cosine decay, an initial learning rate of 0.0005, a weight decay of 0.05, and a batch size of 32 for 300 epochs. Unless specified, we use the same task header as Point-MAE [33] in all downstream tasks. When training from scratch, we use the same settings as in fine-tuning. All experiments are conducted using an NVIDIA RTX 3090 GPU.

### 4.2 Comparison on Downstream Tasks

We show Mamba3D's results on downstream tasks here. For each experiment, we report results for models trained from scratch, as well as those employing two pre-training strategies. Unless specified, the results for Mamba3D do not use a voting strategy.

### 4.2.1 *Object Classification*.
We conduct classification experiments on both the real-world ScanObjectNN [49] dataset and the synthetic ModelNet40 [56] dataset.

***Settings***. ScanObjectNN dataset contains ~15K objects from 15 classes, scanned from the real world with cluttered backgrounds. We experiment with its three variants: OBJ_BG, OBJ_ONLY, and PB_T50_RS. We use *rotation* as data augmentation [10], with a point cloud size $N$=2048. ModelNet40 dataset includes ~12K synthetic

**Table 1: Classification results on the ScanObjectNN and ModelNet40 datasets. The inference model parameters #P (M), FLOPs #F (G), and overall accuracy (%) are reported. We compare with methods using the △ hierarchical Transformer architectures (*e.g.*, Point-M2AE [63]), • plain Transformer architectures, ○ dedicated architectures for 3D understanding, and ★ Mamba-based architectures. † means additional tuning. PT: pre-training strategy.**

| Method | PT | #P ↓ | #F ↓ | ScanObjectNN | | | ModelNet40 |
| | | | | OBJ_BG ↑ | OBJ_ONLY ↑ | PB_T50_RS ↑ | 1k P ↑ |
|---|---|---|---|---|---|---|---|
| *Supervised Learning Only: Dedicated Architectures* | | | | | | | |
| ○ PointNet [35] | × | 3.5 | 0.5 | 73.3 | 79.2 | 68.0 | 89.2 |
| ○ PointNet++ [36] | × | 1.5 | 1.7 | 82.3 | 84.3 | 77.9 | 90.7 |
| ○ DGCNN [52] | × | 1.8 | 2.4 | 82.8 | 86.2 | 78.1 | 92.9 |
| ○ PointCNN [25] | × | 0.6 | - | 86.1 | 85.5 | 78.5 | 92.2 |
| ○ DRNet [41] | × | - | - | - | - | 80.3 | 93.1 |
| ○ SimpleView [14] | × | - | - | - | - | 80.5±0.3 | 93.9 |
| ○ GBNet [42] | × | 8.8 | - | - | - | 81.0 | 93.8 |
| ○ PRA-Net [7] | × | | 2.3 | - - | - | 81.0 | 93.7 |
| ○ MVTN [20] | × | 11.2 | 43.7 | **92.6** | **92.3** | 82.8 | 93.8 |
| ○ PointMLP [31] | × | 12.6 | 31.4 | - | - | 85.4±0.3 | **94.5** |
| ○ PointNeXt [40] | × | 1.4 | 3.6 | - | - | 87.7±0.4 | 94.0 |
| ○ P2P-HorNet [53] | ✓ | - | 34.6 | - | - | 89.3 | 94.0 |
| ○ DeLA [5] | × | 5.3 | 1.5 | - | - | **90.4** | 94.0 |
| *Supervised Learning Only: Transformer or Mamba-based Models* | | | | | | | |
| • Transformer [50] | × | 22.1 | 4.8 | 79.86 | 80.55 | 77.24 | 91.4 |
| △ PCT [18] | × | 2.9 | 2.3 | - | - | - | 93.2 |
| ★ PointMamba [26] | × | 12.3 | 3.6 | 88.30 | 87.78 | 82.48 | - |
| ★ PCM [64] | × | 34.2 | 45.0 | - | - | 88.10±0.3 | 93.4±0.2 |
| △ SPoTr [34] | × | 1.7 | 10.8 | - | - | 88.60 | - |
| △ PointConT [29] | × | - | - | - | - | 90.30 | 93.5 |
| ★ **Mamba3D w/o vot.** | × | 16.9 | 3.9 | **92.94** | **92.08** | **91.81** | 93.4 |
| ★ **Mamba3D w/ vot.** | × | 16.9 | 3.9 | **94.49** | **92.43** | **92.64** | **94.1** |
| *with Self-supervised Pre-training* | | | | | | | |
| • Transformer [50] | *OcCo [51]* | 22.1 | 4.8 | 84.85 | 85.54 | 78.79 | 92.1 |
| • Point-BERT [61] | *IDPT [62]* | 22.1+1.7† | 4.8 | 88.12 | 88.30 | 83.69 | 93.4 |
| • MaskPoint [27] | *MaskPoint* | 22.1 | 4.8 | 89.30 | 88.10 | 84.30 | 93.8 |
| ★ PointMamba [26] | *Point-MAE* | 12.3 | 3.6 | 90.71 | 88.47 | 84.87 | - |
| • Point-MAE [33] | *IDPT [62]* | 22.1+1.7† | 4.8 | 91.22 | 90.02 | 84.94 | 94.4 |
| △ Point-M2AE [63] | *Point-M2AE* | 15.3 | 3.6 | 91.22 | 88.81 | 86.43 | 94.0 |
| • Point-BERT [61] | *Point-BERT* | 22.1 | 4.8 | 87.43 | 88.12 | 83.07 | 93.2 |
| ★ **Mamba3D w/o vot.** | *Point-BERT* | 16.9 | 3.9 | **92.25** *+4.82* | **91.05** *+2.93* | **87.58** *+4.51* | **94.4** *+1.2* |
| • Point-MAE [33] | *Point-MAE* | 22.1 | 4.8 | 90.02 | 88.29 | 85.18 | 93.8 |
| ★ **Mamba3D w/o vot.** | *Point-MAE* | 16.9 | 3.9 | **93.12** *+3.10* | **92.08** *+3.79* | **88.20** *+3.02* | **94.7** *+0.9* |
| ★ **Mamba3D w/ vot.** | *Point-MAE* | 16.9 | 3.9 | **95.18** *+5.16* | **92.60** *+4.31* | **88.97** *+3.79* | **95.1** *+1.3* |

3D CAD models across 40 classes. We use $N$=1024 points as input, and apply *scale&translate* for data augmentation [35].

**Results.** Table 1 reports the comparison results. When trained from scratch, Mamba3D achieved 91.81% overall accuracy (OA) on the most difficult variant PB_T50_RS of ScanObjectNN, and 92.64% after voting, surpassing SoTA model DeLA's 90.4% [5], achieving new SoTA for models trained from scratch. Compared to Transformer [50], Mamba3D gains an OA increase of +15.40%, with only

76% parameters and 81% FLOPs. Notably, Mamba3D surpasses two concurrent works PointMamba [26] and PCM [64] by +10.16% and +4.54%, respectively. On the ModelNet40 dataset, Mamba3D is +2.7% higher than Transformer. Our model surpasses PCM with less than half the parameters (16.9M vs. 34.2M), and only 8.7% FLOPs (3.9G vs. 45.0G).

After pre-training, our proposed Mamba3D consistently outperforms Transformer-based models. With the Point-BERT [61]

**Table 2: Few-shot classification on ModelNet40 dataset. Overall accuracy (%) without voting is reported. *P-B* and *P-M* represent Point-BERT and Point-MAE strategy, respectively.**

| Method | 5-way | | 10-way | |
|---|---|---|---|---|
| | 10-shot ↑ | 20-shot ↑ | 10-shot ↑ | 20-shot ↑ |
| *Supervised Learning Only* | | | | |
| ○ DGCNN [52] | 31.6 ± 2.8 | 40.8 ± 4.6 | 19.9 ± 2.1 | 16.9 ± 1.5 |
| ● Transformer [50] | 87.8 ± 5.2 | 93.3 ± 4.3 | 84.6 ± 5.5 | 89.4 ± 6.3 |
| ★ Mамва3D | **92.6** ± 3.7 | **96.9** ± 2.4 | **88.1** ± 5.3 | **93.1** ± 3.6 |
| *with Self-supervised Pre-training* | | | | |
| ○ DGCNN+*OcCo* | 90.6 ± 2.8 | 92.5 ± 1.9 | 82.9 ± 1.3 | 86.5 ± 2.2 |
| ● OcCo [51] | 94.0 ± 3.6 | 95.9 ± 2.7 | 89.4 ± 5.1 | 92.4 ±4.6 |
| ★ PointMamba [26] | 95.0 ± 2.3 | 97.3 ± 1.8 | 91.4 ± 4.4 | 92.8 ± 4.0 |
| ● MaskPoint [27] | 95.0 ± 3.7 | 97.2 ± 1.7 | 91.4 ± 4.0 | 93.4 ± 3.5 |
| ● Point-BERT [61] | 94.6 ± 3.1 | 96.3 ± 2.7 | 91.0 ± 5.4 | 92.7 ± 5.1 |
| ★ Mамва3D+*P-B* | **95.8** ± 2.7 | **97.9** ± 1.4 | **91.3** ± 4.7 | **94.5** ± 3.3 |
| ● Point-MAE [33] | 96.3 ± 2.5 | 97.8 ± 1.8 | **92.6** ±4.1 | 95.0 ± 3.0 |
| ★ Mамва3D+*P-M* | **96.4** ± 2.2 | **98.2** ±1.2 | 92.4 ± 4.1 | **95.2** ± 2.9 |

**Table 3: Part segmentation on ShapeNetPart dataset. The class mIoU (mIoU$_C$) and the instance mIoU (mIoU$_I$) are reported, with model parameters #P (M) and FLOPs #F (G).**

| Method | mIoU$_C$ (%) ↑ | mIoU$_I$ (%) ↑ | #P ↓ | #F ↓ |
|---|---|---|---|---|
| *Supervised Learning Only* | | | | |
| ○ PointNet [35] | 80.4 | 83.7 | 3.6 | 4.9 |
| ○ PointNet++ [36] | 81.9 | 85.1 | **1.0** | 4.9 |
| ○ DGCNN [52] | 82.3 | 85.2 | 1.3 | 12.4 |
| ● Transformer [50] | 83.4 | 85.1 | 27.1 | 15.5 |
| ★ Mамва3D | **83.7** | **85.7** | 23.0 | 11.8 |
| *with Self-supervised Pre-training* | | | | |
| ● OcCo [51] | 83.4 | 84.7 | 27.1 | - |
| ○ PointContrast [57] | - | 85.1 | 37.9 | - |
| ○ CrossPoint [1] | - | 85.5 | - | - |
| ● Point-BERT [61] | 84.1 | 85.6 | 27.1 | 10.6 |
| ★ Mамва3D+*P-B* | 84.1 | 85.7 | **21.9** | 9.5 |
| ● Point-MAE [33] | 84.2 | **86.1** | 27.1 | 15.5 |
| ★ PointMamba [26] | **84.4** | 86.0 | **17.4** | 14.3 |
| ★ Mамва3D+*P-M* | 83.6 | 85.6 | 23.0 | **11.8** |

strategy, Mамва3D surpasses Point-BERT by +4.51% on ScanObjectNN and +1.2% on ModelNet40, also outperforming hierarchical Transformer model Point-M2AE [63] by +1.15% and +0.4% on this two datasets. When using the Point-MAE [33] strategy, Mамва3D achieves 95.1% on ModelNet40, setting new SoTA for single-modal pre-trained models. On the ScanObjectNN dataset, Mамва3D outperforms Transformer with OcCo [51] by +10.2%, and Point-MAE by +3.8%. Besides, we gained an increase of +4.1% compared to Point-Mamba [26] with the same pre-training strategy. Overall, these results highlight Mамва3D's superiority over existing dedicated architectures and Transformer- or Mamba-based models, achieving multiple SoTA, demonstrating its strength across various settings.

*4.2.2* ***Few-shot Learning***. We conduct few-shot classification experiments following previous work [46], to further validate few-shot learning ability of Mамва3D.

**Settings**. We use ModelNet40 dataset [56] with an *n*-way, *m*-shot setting, where *n* is the number of classes randomly sampled from the dataset, and *m* denotes the number of samples randomly drawn from each class. We train the model with only the sampled $n \times m$ samples. During testing, we randomly select 20 novel objects for each of the *n* classes to serve as test data. We experiment with $n \in \{5, 10\}$ and $m \in \{10, 20\}$. For each setting, we report the mean accuracy and standard deviation of 10 independent experiments.

**Results**. Table 2 reports the comparison results. When trained from scratch, Both Mамва3D and Transformer significantly surpass DGCNN [52] by a large margin. Mамва3D outperforms Transformer [50] with overall accuracy (OA) improvements of +4.8%, +3.6%, +3.5%, and +3.7% across four settings, respectively, with also smaller deviations and fewer FLOPs. Under the Point-BERT strategy, Mамва3D outperformed Point-BERT [61] by +1.2%, +1.6%, +0.3%, and +1.8%, respectively, and with smaller deviations. Similarly, with the Point-MAE [33] strategy, Mамва3D outperforms Point-MAE on three out of four settings, and surpasses PointMamba [26] on

all settings. These few-shot experiments demonstrate Mамва3D's adeptness at learning semantic information and its efficient knowledge transfer ability to downstream tasks, even with limited data.

*4.2.3* ***Part Segmentation***. We conducted part segmentation on the ShapeNetPart dataset [59] to predict more fine-grained class labels for every point.

**Settings**. ShapeNetPart dataset comprises ~16K objects across 16 categories. We use a segmentation head similar to Point-BERT [61]-utilizing PointNet++ [36] in feature propagation, along with a similar feature extraction strategy—employing features from the 4th, 8th, and 12th encoder layers. We use input point cloud *N*=2048 without normal, and employ cross-entropy as the loss function.

**Results**. Table 3 reports the comparison results in terms of the average instance IoU (mIoU$_I$) and average category IoU (mIoU$_C$). With supervised training alone, Mамва3D surpasses Transformer by +0.3% in mIoU$_C$ and +0.6% in mIoU$_I$. With the Point-BERT strategy, Mамва3D achieves +0.1% higher mIoU$_I$ compared to Point-BERT. While Mамва3D yields slightly lower results than Point-MAE, it employs 17.8% fewer parameters and 31.4% fewer FLOPs. The segmentation experiments further demonstrate the effectiveness and efficiency of Mамва3D.

## 4.3 Ablation Study

We conduct ablation studies on model structure and also investigate the effect of ordering strategies. We report the results of training from scratch on the ScanObjectNN (OBJ_ONLY) dataset.

*4.3.1* ***Architecture Ablation***. Results of ablation on architecture are shown in Table 4. Removing the LNP block (w/o LNP) and the bi-SSM block (w/o bi-SSM) separately results in a 1.2% and 2.3% degradation in overall accuracy (OA), respectively. To further validate the effect of the bi-SSM block, we design four variants. Firstly, we replace it with a self-attention [50] layer (LNP+Attn),

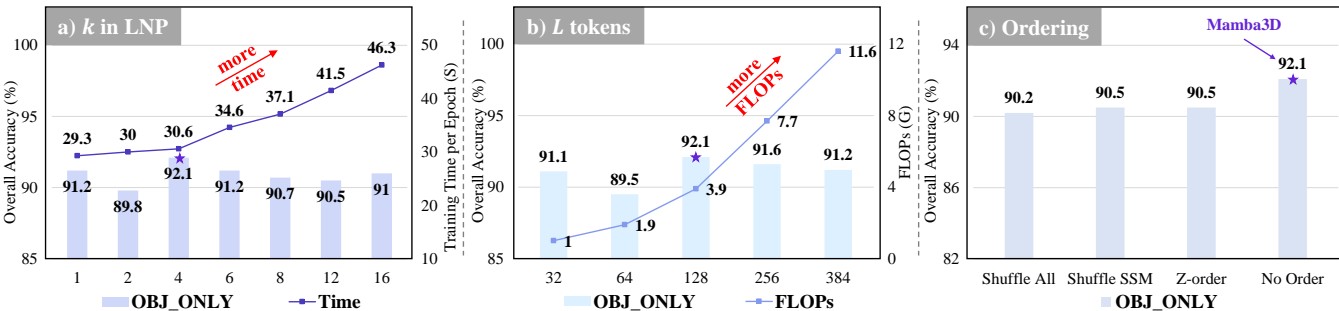

**Figure 4: Ablation on (a) the parameter $k$ in the LNP block, (b) input patch sequence length $L$, and (c) ordering strategy. The overall accuracy (%), training time/epoch (s) and FLOPs (G) are reported.**

**Table 4: Ablation on model architecture.**

| Method | OBJ_ONLY (%) ↑ | Params (M) ↓ | FLOPs (G) ↓ |
|---|---|---|---|
| ★ **Full** | **92.1** | 16.9 | 3.9 |
| w/o LNP | 90.9 *-1.2* | 13.3 | 3.4 |
| w/o bi-SSM | 89.8 *-2.3* | 4.4 | 2.5 |
| ★ tri-SSM | 91.0 *-1.1* | 17.9 | 3.9 |
| ★ one-SSM | 90.9 *-1.2* | 16.0 | 3.9 |
| ● LNP + Attn | 90.9 *-1.2* | 25.7 | 5.4 |
| ★ Token Flip | 90.7 *-1.4* | 16.9 | 3.9 |

**Table 5: Ablation on K-norm.**

| Method | OBJ_ONLY (%) ↑ | Params (M) ↓ | FLOPs (G) ↓ |
|---|---|---|---|
| **K-norm** | **92.1** | 16.93 | 3.86 |
| w/o rela. dist. | 91.6 *-0.5* | 16.93 | 3.86 |
| w/ K linear | 90.9 *-1.2* | 16.98 | 3.86 |
| w/o linear | 90.5 *-1.6* | 16.91 | 3.86 |
| w/o K-norm | 89.2 *-2.9* | 21.25 | 5.98 |
| w/o $F_C$ | 88.8 *-3.3* | 15.14 | 3.63 |
| w/o concat. | 88.6 *-3.5* | 15.14 | 3.63 |

which leads to a 1.2% reduction in OA. When using only a unidirectional SSM (one-SSM), the OA decreases by 1.2%. Exploring token flip as an alternative to the channel flip (Token Flip) results in a 1.4% OA degradation, and directly adding a token flip in bi-SSM block (tri-SSM) leads to a 1.1% drop. These results demonstrate the effectiveness of the C-SSM block for unordered points.

*4.3.2 K-norm Ablation.* Results of ablation on K-norm are shown in Table 5. Removing K-norm entirely (w/o K-norm) decreases OA by 2.9%. We extensively verify the effectiveness of the K-norm operator defined in Eq. (5). Dropping the concatenated $F_C$ (w/o concat.) lowered OA to 88.6%, and removing $F_C$ completely (w/o $F_C$) decreases OA by 3.3%. Without linear transformation (w/o linear), OA decreases by 1.6%, and using distinct linear transformations for $K$ neighbors (K linear) leads to a 1.2% drop, underscoring K-norm's simplicity and efficacy. Even without the centralizing $F_C$, the model achieved 91.6% OA. These results highlight the K-norm's role in effectively transmitting local information and improving local geometric capture.

*4.3.3 K-pooling Ablation.* More detailed ablation results for K-pooling are shown in Table 6. Replacing K-pooling with one MLP

**Table 6: Ablation on K-pooling.**

| Method | OBJ_ONLY (%) ↑ | Params (M) ↓ | FLOPs (G) ↓ |
|---|---|---|---|
| **K-pooling** | **92.1** | 16.93 | 3.86 |
| w/ Maxpool | 91.4 *-0.7* | 16.93 | 3.86 |
| w/o K-pool | 89.8 *-2.3* | 17.72 | 4.47 |
| w/ Max + Avgpool | 89.7 *-2.4* | 16.93 | 3.86 |
| w/ Avgpool | 89.3 *-2.8* | 16.93 | 3.86 |

(w/o K-pool) leads to an increase of +0.8M in parameters and +0.61G in FLOPs, while the OA decreases by 2.3%, highlighting the simplicity and effectiveness of K-pooling. When K-pooling is substituted with Avgpooling, Maxpooling, or Max+Avgpooling, the OA is reduced by 2.8%, 0.7%, and 2.4%, respectively, indicating the efficacy of the simple K-pooling operator in aggregating local features.

*4.3.4 Parameters and Ordering.* We also analyze the model's performance under various model parameters, as depicted in Fig. 4(a-b). In Fig. 4(a), we adjust the $k$ in LNP to change the size of the local patch graph. Results show that as the local neighborhood increases, so does the model training time, with the overall accuracy (OA) reaching its peak at $k$=4. Results in Fig. 4(b) indicate that as the token length $L$ increases, so do the FLOPs, with the OA peaking at 92.1% when $L$=128. Lastly, in Fig. 4(c), we investigate ordering strategies and find that MAMBA3D performs optimally without any ordering strategy applied. This suggests that MAMBA3D effectively captures the semantic features from unordered points.

## 5 CONCLUSION

We introduce MAMBA3D, a novel SSM-based architecture for point cloud learning. MAMBA3D surpasses Transformer-based models across various tasks while maintaining linear complexity. Specifically, our LNP block, comprising K-norm and K-pooling, facilitates the explicit local feature injection. Also, we propose C-SSM to adapt the SSM for better handling unordered points. Through extensive experimentation and validation, MAMBA3D achieves multiple SoTA across multiple tasks with only linear complexity. We aspire for MAMBA3D to advance the field of large point cloud models.

*Discussion.* There are still limitations to this work. Pre-training bonus is not as robust as the Transformer, possibly due to unsuitable masked point modeling for recurrent models like Mamba. We will explore tailored pre-training strategies and scale up the model to optimize its linear complexity advantage in future research.

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
