# OpenReview forum: "Mamba3D: Enhancing Local Features for 3D Point Cloud Analysis via State Space Model"
_acmmm.org/ACMMM/2024/Conference — MM2024 Poster_

### Official Review · Reviewer_9H84 · 2024-05-07

**Rating:** 5
**Confidence:** 4

**Summary:**

The paper introduces a novel state space model, Mamba3D, for point cloud analysis, addressing the limitations of Transformer-based models.

**Strengths:**

The authors leverage SSMs to overcome the quadratic bottleneck of Transformers and achieve efficient point cloud feature learning. They introduce a novel selection mechanism and hardware-accelerated scan in Mamba, enabling near-linear complexity during training.

The paper provides a clear and well-structured explanation of the proposed model and its components, including LNP for local feature extraction.

**Limitations:**

Further evaluation of larger datasets and real-world scenarios could enhance the paper's credibility.

**Suitability:**

3

---

### Official Review · Reviewer_PhGw · 2024-05-19

**Rating:** 4
**Confidence:** 3

**Summary:**

Mamba3D incorporates a Local Norm Pooling (LNP) block to effectively capture local geometric features. Additionally, it employs a bidirectional SSM (bi-SSM) that includes both a token forward SSM and a novel backward SSM operating on the feature channel to enhance global feature representation. The proposed method achieves superior performance compared to Transformer-based models and other contemporary methods, demonstrating its efficacy through extensive experimental results.

Noteworthy achievements of Mamba3D include setting new state-of-the-art results on several tasks: an overall accuracy of 92.6% on the ScanObjectNN classification task when trained from scratch, and 95.1% on the ModelNet40 classification task with single-modal pre-training, all while maintaining linear complexity. The paper concludes with a commitment to release the code and model upon publication, further contributing to the research community.

**Strengths:**

1. Novelty and Innovation: The introduction of Mamba3D is a significant advancement in the field of point cloud analysis. The model leverages state space models (SSM) to address the limitations of Transformer-based models, particularly their quadratic complexity. This innovative approach not only reduces computational complexity to linear but also enhances the extraction of local features through the proposed Local Norm Pooling (LNP) block, which is a simple yet effective mechanism for capturing local geometric information.

2. Comprehensive Evaluation and Empirical Validation: The authors have conducted extensive experiments to validate the effectiveness of Mamba3D. The model's performance is thoroughly evaluated on multiple benchmark datasets, including ScanObjectNN and ModelNet40. The results demonstrate that Mamba3D surpasses existing Transformer-based and other state-of-the-art methods in terms of accuracy, with notable achievements such as 92.6% accuracy on ScanObjectNN and 95.1% on ModelNet40 with pre-training. This robust evaluation underlines the model's technical correctness and applicability to real-world tasks.

3. Clarity and Reproducibility: The paper is well-organized and clearly presents the proposed methodologies, theoretical underpinnings, and experimental results. The authors' commitment to releasing the code and model upon publication further enhances the paper's value, as it allows other researchers to reproduce and build upon their work, fostering further advancements in the field.

**Limitations:**

Lack of Evaluation on Large-Scale Point Cloud Benchmarks: While the paper demonstrates Mamba3D's effectiveness on smaller-scale datasets like ScanObjectNN and ModelNet40, it does not provide experimental results on large-scale point cloud benchmarks such as ScanNet and S3DIS. Given that one of the key advantages of Mamba model over Transformer-based models is its linear complexity, showcasing its performance on these large-scale datasets is crucial to fully validate the scalability and practical benefits of the proposed method. Without these results, the claimed superiority of Mamba3D in point cloud analysis is unconvincing.

**Suitability:**

2

---

### Official Review · Reviewer_A9qg · 2024-05-24

**Rating:** 4
**Confidence:** 2

**Summary:**

This article introduces Mamba3D, a novel 3D point cloud analysis architecture based on the State Space Model (SSM). By incorporating innovative Local Norm Pooling (LNP) blocks and a bidirectional SSM (bi-SSM) design, it effectively enhances local feature extraction and addresses the challenges of unordered point clouds. It achieves superior performance and linear complexity in multiple downstream tasks, although further research and improvement are needed in pre-training strategies and model generalization capabilities.

**Strengths:**

**Advantages:**
1. **Performance and Efficiency:** Mamba3D outperforms Transformer-based models in terms of both accuracy and computational efficiency. It operates with linear complexity, which is beneficial for processing large datasets.
2. **Enhanced Feature Extraction:** The introduction of Local Norm Pooling (LNP) and a bidirectional state space model (bi-SSM) enhances the local and global feature extraction capabilities of the system, leading to improved performance in tasks like 3D object detection and segmentation.
3. **Scalability and Adaptability:** The model shows strong scalability and adaptability across various tasks, evidenced by its superior performance in both supervised and few-shot learning settings.

**Limitations:**

**Disadvantages:**
1. **Computational Resources:**  Although Mamba3D has linear complexity, the actual consumption of computational resources, such as GPU memory and training time, is not discussed in detail.
2. **Dependence on Quality of Training Data:** The effectiveness of Mamba3D is somewhat dependent on the quality and arrangement of the training data, particularly when dealing with actual collected point cloud data, such as kitti and nuScenes dataset.
3. **Limited by Pre-training Strategies:** The model's performance is significantly boosted by pre-training, suggesting it might not perform as well without these initial adjustments, which could be a limitation in environments where pre-training is not feasible.

**Suitability:**

3

---

### Meta-Review · Area_Chair_j1Qz · 2024-07-03

**Recommendation:** Accept (Poster)
**Confidence:** 5

**Metareview:**

This paper was reviewed by three experts in the field. The recommendations are (Weak Accept, Borderline Accept x 2). Based on the reviewers' feedback, the decision is to recommend the acceptance of the paper. The reviewers did raise some valuable concerns (especially further experimental evaluations on larger datasets, benchmarks, and real-world scenarios) that should be addressed in the final camera-ready version of the paper. The authors are encouraged to make the necessary changes to the best of their ability. We congratulate the authors on the acceptance of their paper.